

# Acoustic velocity measurements for detecting the crystal orientation fabrics of a temperate ice core

Sebastian Hellmann[1,2], Melchior Grab[1,2], Johanna Kerch[3,4], Henning Löwe[5], Andreas Bauder[1], Ilka Weikusat[3,6], and Hansruedi Maurer[2]

[1]Laboratory of Hydraulics, Hydrology and Glaciology (VAW), ETH Zurich, Zurich, Switzerland
[2]Institute of Geophysics, ETH Zurich, Zurich, Switzerland
[3]Alfred-Wegener-Institut Helmholtz-Zentrum für Polar- und Meeresforschung, Bremerhaven, Germany
[4]GZG Computational Geoscience, Georg-August University, Göttingen, Germany
[5]WSL Institute for Snow and Avalanche Research SLF, Davos, Switzerland
[6]Department of Geosciences, Eberhard Karls University, Tübingen, Germany

**Correspondence:** Sebastian Hellmann (sebastian.hellmann@erdw.ethz.ch)

**Abstract.** The crystal orientation fabrics (COF) in ice cores provides detailed information, such as grain size and distribution and the orientation of the crystals in relation to the large-scale glacier flow. These data are relevant for a profound understanding of the dynamics and deformation history of glaciers and ice sheets. The intrinsic, mechanical anisotropy of the ice crystals causes an anisotropy of the polycrystalline ice of glaciers and affects the velocity of acoustic waves propagating through the

ice. Here, we employ such acoustic waves to obtain the seismic anisotropy of ice core samples and compare the results with calculated acoustic velocities derived from COF analyses. These samples originate from an ice core from Rhonegletscher, a temperate glacier in the Swiss Alps. Point-contact transducers transmit ultrasonic p-waves with a dominant frequency of 1 MHz into the ice core samples and measure variations of the travel times of these waves for a set of azimuthal angles. In addition, the elasticity tensor is obtained from laboratory-measured COF and calculate the associated seismic velocities. We

compare these COF-derived velocity profiles with the measured ultrasonic profiles. Especially in the presence of large ice grains, these two methods show significantly different velocities since the ultrasonic measurements examine a limited volume of the ice core whereas the COF-derived velocities are integrated over larger parts of the core. This discrepancy between the ultrasonic and COF-derived profiles decreases with an increasing number of grains that is available within the sampling volume and both methods provide concise results in presence of a similar amount of grains. We also explore the limitations

of ultrasonic measurements and provide suggestions for improving their results. These ultrasonic measurements could be employed continuously along the ice cores. They are suitable to support the COF analyses by bridging the gaps between discrete measurements, since these ultrasonic measurements can be acquired within minutes and do not require an extensive preparation of ice samples when using point-contact transducers.

## 1 Introduction

Improved glacier flow models require a profound knowledge on sub- and englacial processes and the properties governing these processes. The data for studying englacial processes are usually derived either from borehole measurements or from ice core





analyses. These ice core analyses provide useful physical properties, such as elastic parameters, density, electric conductivity and permittivity (e.g. Freitag et al., 2004; Wilhelms, 2005). Most of these properties are anisotropic in ice cores, because the physical properties of a single ice crystal vary along its principal axes, and the crystals usually exhibit preferential orientations

under deformation. Furthermore, ice cores provide geometric details on the ice microstructure, such as grain size and shape, and information on crystal orientation. The derived crystal orientation fabric (COF) describes the orientation of the ice grains' c-axes, which are the symmetry axes of the individual ice monocrystals in the polycrystalline material. The COF is governed by the stress field and the ice deformation and thus preserves the ice flow history of a glacier or ice sheet (Budd, 1972; Azuma and Higashi, 1984; Alley, 1988). It is also an indicator for the internal ice structure at the ice core location, which allow a

classification of the ice as "soft" or "hard" depending on the direction of the strain rates relative to the COF (e.g. Budd and Jacka, 1989; Faria et al., 2014). Such information is crucial for improved glacier flow models that consider anisotropy effects (Alley, 1992; Azuma, 1994). For example, information on the anisotropic ice flow dynamics of a glacier have successfully been incorporated in ice flow models by Gillet-Chaulet et al. (2005), Placidi et al. (2010), and Graham et al. (2018).

For the analysis of the COF, thin sections of ice ($\approx 350\,\mu$m thick) are manually prepared from ice core samples and finally

measured with an automated fabric analyser (e.g. Wilson et al., 2003; Peternell et al., 2009). This processing workflow is state-of-the-art, but it is labour-intensive and usually yields only discrete measurements along the entire ice core. Therefore, other methods have been proposed in the last decades. Initial attempts to develop new methods were conducted throughout the late 1980's and early 1990's. Langway et al. (1988) developed a tool that uses p-waves for COF detection. This methodology required a preparation of the core samples to obtain plane-parallel surfaces on which the plain transducers could be attached.

Later, Anandakrishnan et al. (1994) advanced this methodology by developing a concept with shear waves that reduced the labour-intensive preparation of the core samples. Recently, Gerling et al. (2017) used travel time differences of acoustic waves to determine the elastic modulus of snow. During the past years, modern non-contacting laser ultrasound acquisition systems have been developed for different purposes, such as investigating stratigraphic layering of ice cores for dating (Mikesell et al., 2017). These methods investigate the elastic parameters of the ice, but can also be employed for COF analyses, since elastic

parameters and COF are directly related.

Important factors to consider when designing a measurement procedure for COF analyses are grain size and shape of the ice samples or the air bubble distribution, which influences the density of the ice. The grain size and shape differ significantly between cold ice and temperate ice. Cold ice typically has larger quantities of small (millimetre-sized) grains, whereas temperate ice has significantly fewer grains, but they are larger with their diameter being up to several centimetres. Furthermore, the

grains in temperate ice are often more irregularly shaped and interlocked and consequently appear as several individual grains within the thin sections. This often leads to a misinterpretation of the actual COF (Budd, 1972; Hooke and Hudleston, 1980; Monz et al., 2020). The large grain size in temperate ice may also affect the afore-mentioned ultrasonic measurements as a result of fewer grains within the ice core volume.

Different geophysical methods have been employed to explore the horizontal extension of the major layers of changing COF

(e.g Bentley, 1975; Blankenship and Bentley, 1987; Matsuoka et al., 2003; Drews et al., 2012; Diez et al., 2015; Picotti et al., 2015). Surface geophysical methods provide easy access to the dominant COF layers in ice sheets (Brisbourne et al., 2019;



Jordan et al., 2019). For more detailed investigations, borehole sonic experiments (Bentley, 1972; Pettit et al., 2007; Gusmeroli et al., 2012) are suitable methods to analyse the COF in a (sub-)metre range, but they have not been considered in recent ice core projects. For all these experiments, the seismic velocity is considered as benchmark. Therefore, Maurel et al. (2015) advanced

a theoretical approach of Nanthikesan and Sunder (1994) to calculate the seismic velocities from given COF to compare them with directly measured sonic velocities in boreholes. Diez and Eisen (2015) developed a very similar theoretical framework to calculate the expected seismic velocities for given COF pattern (including cone, thick and partial girdle fabrics as they are typically found in polar ice cores) for comparison with surface geophysical investigations. This was extended to a more general framework by Kerch et al. (2018) for any given COF.

A direct comparison of the measured and calculated velocities is still limiting as the measured data may be affected by macro-structural features such as crevasses, fractures, changing ice porosity due to air bubbles or meltwater within the ice matrix. In order to avoid all these limitations, direct ultrasonic measurement along an ice core, from which the COF is usually derived, could be employed and may provide the best agreement between COF-derived and measured acoustic velocities. Such a comparison of ultrasonic measurements with COF-derived velocities ist the aim of this study. We obtain seismic velocities

from ultrasonic measurements on ice core samples from the temperate Rhonegletscher in Switzerland. We already analysed the actual COF of these ice core samples in a recent study (Hellmann et al., 2020b) and use the framework of Kerch et al. (2018) to calculate the COF-derived seismic velocity profiles. CT analyses are incorporated to account for air bubbles that are still affecting this comparison. We demonstrate the potential of our ultrasonic method applied to an ice core to directly link to the complementary fabric measurements acquired with polarisation microscopy and provide suggestions for further improvements.

To our knowledge, this is the first comparative study of COF-derived and ultrasonic velocity analyses on temperate ice.

## 2 Data acquisition and Methods

### 2.1 Ice core fabric data

For our velocity investigations we used an ice core drilled on Rhonegletscher, Central Swiss Alps. The ice core was drilled in August 2017 with a recently developed thermal drilling technique suitable for temperate ice (Schwikowski et al., 2014). An

80 m long ice core was retrieved in the ablation area of the glacier (N46° 35.220' / E008° 23.268'; 2314 m a.s.l. in 2017). In location of the drilled ice core, the glacier flows from northwestern ($\approx 335° \pm 10°$) to southeastern direction.

Immediately after extracting the ice core, it was stored at -30°C. This caused any water-filled pores within the ice matrix to freeze. Seven samples (with 0.5 m length each) along the ice core were analysed at Alfred Wegener Institute Helmholtz Centre for Polar and Marine Research (AWI) Bremerhaven in order to obtain a comprehensive crystal orientation fabric (COF)

dataset. For each sample, eight to twelve horizontal and vertical thin sections (covering three perpendicular planes) from two adjacent ice core segments were prepared. The COF was then analysed with polarised light microscopy (Peternell et al., 2009). We employed the automatic fabric analyser G50 from Russell-Head Instruments (e.g. Wilson et al., 2003) and the software *cAxes* (Eichler, 2013) to obtain a comprehensive fabric dataset for each ice core sample. The results of this COF analysis are presented in Figs. 2a to 2g and further details can be found in Hellmann et al. (2020b).





## 2.2 Seismic velocities from COF


The hexagonal crystal structure of an ice monocrystal causes an anisotropy in its elastic parameters and therefore affects the propagation velocity of seismic waves. As a result of the crystallographic symmetry, the acoustic velocity parallel to the c-axis, which corresponds to the optical axis perpendicular to the basal planes of the ice crystal lattice (see e.g. Cuffey and Paterson, 2010), differs significantly from the velocity in direction of the basal plane. This seismic anisotropy of an ice crystal is fully

described by the fourth order elasticity tensor $C_{ijkl}$, i,j,k,l = 1, 2, 3 (e.g. Aki and Richards, 2002). The velocity of an acoustic wave with any inclination and azimuthal direction can be calculated analytically (Tsvankin, 2001), provided the mass density of ice is known.

Due to the symmetry relations (Voigt, 1910) the 81 unknown elements of the tensor can be reduced to 21 elements. The hexagonal symmetry of ice further reduces the number of independent constants to five for a monocrystal. For the determination

of a representative elasticity tensor for a polycrystalline medium, we follow the approach of Kerch et al. (2018). A detailed description on calculating the polycrystalline tensor can be found there.

The theoretical framework calculates the effective elasticity tensor and derives the seismic velocities from this tensor. Then, the velocities are derived by solving the Christoffel Equation (e.g. Tsvankin, 2001, Ch. 1.1.2). According to Maurel et al. (2016), this approach for an effective elasticity tensor provides more accurate results (at least for some specific textures) than the

complementary velocity averaging method (i.e. calculating the velocities for the individual crystals and computing the average velocity for the polycrystalline medium afterwards). Here, we only summarise the key points for calculating the effective elasticity tenor.

- This approach is based on an earlier study of Diez and Eisen (2015). However, the framework of Diez and Eisen (2015) relies on particular COF patterns, such as a thick and partial girdle or a single maximum structure and their representation

through the eigenvalues of the orientation tensor. Kerch et al. (2018) do not presume specific COF patterns, which makes it most suitable for our dataset.

- It then considers the elements of a monocrystal tensor $C_{\mathrm{m}}$ precisely determined in laboratory experiments. In our study, we used the elasticity tensor of Bennett,

$$
C_{\mathrm{m}} = \begin{bmatrix}
14.06 & 7.15 & 5.88 & 0 & 0 & 0 \\
7.15 & 14.06 & 5.88 & 0 & 0 & 0 \\
5.88 & 5.88 & 15.24 & 0 & 0 & 0 \\
0 & 0 & 0 & 3.06 & 0 & 0 \\
0 & 0 & 0 & 0 & 3.06 & 0 \\
0 & 0 & 0 & 0 & 0 & 3.455
\end{bmatrix} \times 10^9 \, \mathrm{N\,m^{-2}},
$$

115 calculated by Bennett (1968) for T = -10°C. This provides the best agreement between our calculated and measured data in both our study and in earlier experiments (Diez et al., 2015, Tab. 1).





- For each ice grain $i$ the monocrystal tensor $C_{\mathrm{m}}$ is transformed into

$$C_{\mathrm{m}}^{\mathrm{rot}}(i) = R_{\vartheta}^{T}(i) R_{\varphi}^{T}(i) C_{\mathrm{m}} R_{\varphi}(i) R_{\vartheta}(i),$$

  where $R_{\vartheta}$ is the rotational matrix around the vertical axis, $R_{\varphi}$ is the rotational matrix around geographic north and $\vartheta$ and $\varphi$ are azimuth and colatitude angle of the grain i, respectively. This aligns the elasticity tensor with the coordinate system of the ice core (transformation of the coordinate system).

- The rotated monocrystal tensors are summed up elementwise

$$C_{\mathrm{p}} = \sum_{i=1}^{n_{\mathrm{G}}} w_{\mathrm{G}}(i)\, C_{\mathrm{m}}^{\mathrm{rot}}(i)$$

  thereby assuming a superposition of all $n_{\mathrm{G}}$ grains and their respective properties. The relative grain sizes are used as weighting factors $w_{\mathrm{G}}(i)$ for each grain. The resulting polycrystalline tensor does not have a hexagonal structure anymore, but a triclinic structure with 21 independent elements.

- With the known elastic properties, the Christoffel Equation provides the link to analytic solutions for acoustic velocities $v_{\mathrm{p}}$, $v_{\mathrm{SH}}$, and $v_{\mathrm{SV}}$, described in detail in Maurel et al. (2015) and Kerch et al. (2018).

We calculated the seismic velocities from the stress and the inverse compliance tensor to obtain the upper and lower bounds of the potential velocity range (also known as Voigt and Reuss bounds) and the mean velocity (Hill, 1952). This analytic solution is in agreement with the numerical approach implemented in the MATLAB Toolbox MTEX (e.g. Mainprice et al., 2011) for crystallographic applications. As the elasticity tensor had been measured at -10°C, we implemented a temperature correction of -2.3 m s$^{-1}$ K$^{-1}$ after Kohnen (1974) to compare the calculated velocities with the ultrasonic measurements at -5°C or in-situ seismic data ($\approx$ -0.5°C).

## 2.3 Ultrasonic experiments on ice core samples

The dominating COF causes an acoustic velocity anisotropy and this anisotropy can be verified and quantified by direct laboratory measurements. These measurements were conducted in the cold laboratory at WSL Institute for Snow and Avalanche Research (SLF) Davos.

The orientation of each individual ice core segment was marked at the time of drilling, based on mechanical onsets and supporting magnetometric measurements. This ensures a comparison between COF, ultrasonic measurements, and glacier flow at all depths. The temperature for the ultrasonic measurements was chosen to be at -5°C. This is a compromise between temperate ice conditions and a controlled cold environment in order to avoid melting effects during the measurement.

An ultrasonic point-contact transducer transmitted an acoustic signal into the ice. This signal was recorded by a second transducer on the opposite side of the core. In the current experimental setup only measurements parallel and perpendicular to the vertical axis of the ice core (colatitude $\varphi = 0/90°$) were considered. The azimuthal coverage for $\varphi = 90°$ was $\Delta \vartheta = 15°$ between 0–345°.



Figure 1a shows the experimental setup that consists of a pulse generator, an oscilloscope, and a set of point contact transducers. The pulse generator (LeCroy wave station) was employed to generate a pulse with a dominant frequency of 1 MHz
and a repetition rate of 10 ms. This electric signal was amplified (amplifiers not shown in Fig. 1a) and a point contact (PC) transducer converted it into an acoustic signal and transmitted it into the ice. This transducer was manufactured in-house at ETH Zurich and provides a stable and highly repeatable sources over a wide range of radiation angles due to its broadband instrument response. This instrument response was calculated in advance using the capillary fracture methods described in Selvadurai (2019). A second transducer (type KRNBB-PC) received and converted the acoustic signal into an electronic pulse,
which was transferred to a digital oscilloscope (LeCroy Wavesurfer 3024). For each measurement, we stacked at least 20 individual waveforms to enhance the signal-to-noise ratio. Since the amplifiers caused delays, we determined the actual zero-time of the entire system by a regression through repeated measurements on steel cylinders with precisely determined lengths. These calibration measurements were performed at least twice a day under identical temperature conditions.

The transducers were screwed in an aluminium tube which was held by an aluminium frame with an inner diameter of 90 mm
in which the ice core with a variable diameter (average diameter: 68±0.36 mm) was placed. In addition to the horizontal measurements, vertical measurements were performed (average length of the samples: 70.6±1.3 mm). The 1 MHz source pulse generated signals with wavelengths of $\approx 3.8$ mm. This resulted in a sample size to wavelength ratio of approximately 20. Thus the wavelength is small enough to measure an integrated velocity as superposition of the individual grain velocities. However, as a result of the large ice grains present in temperate ice, some measurements maybe biased by only a few larger grains. We
performed measurements at three levels of the ice core samples (denoted as $z_i$, $i = 1, 2, 3$ in Fig. 1a, offset $\Delta z \approx 35$ mm) and averaged the results. We assume this stacking procedure to be comparable with the combination of several thin sections for the COF analysis.

## 2.4   X-ray measurements for air content estimation

In addition to the ultrasonic measurements, the porosity (i.e. the volume of air within the ice) was analysed by X-ray micro-
computer tomography (CT) scans. For the scanning and analysis, we followed the same procedures previously adopted for bubbly ice from Dome C (Fourteau et al., 2019). The samples placed within the CT scanner had a diameter of approximately 18 mm and a length of 70 mm resulting in images with a resolution (voxel size) of $(10\,\mu m)^3$ (Gerling et al., 2017). A set of 2-3 regions of interest (ROIs) with a maximum height of 15 mm each was defined in the vicinity of the horizontal levels of ultrasonic measurements, which were about 35 mm apart. The grey scale images of the ROIs were automatically segmented
into binary (ice/air) images following the method from Hagenmuller et al. (2013). The air volume fraction was subsequently calculated from the binary images as the fraction of air voxels in the image. The ice core consists mainly of ice and air captured in bubbles. Liquid water was refrozen during storage of the core segments and dust and sediment particles can be neglected. Therefore, we classify the images as two-phase systems with air bubbles in ice.





## 3 Data analysis and results

### 3.1 Acoustic velocities inferred from COF

Seven ice core samples, obtained from 2, 22, 33, 45, 52, 65 and 79 m depth, were analysed. The corresponding COF patterns (presented in Figs. 2a – 2g) are obtained from a set of ice core thin sections from two adjacent ice core segments. They exhibit clear multi-maxima patterns in all samples, consisting of four (five for 65 m) significant clusters of c-axes. These clusters always form a "diamond shape" pattern and have been found to be typical for temperate ice with branched, large ice grains. We employed a spherical k-means clustering algorithm (Nguyen, 2020) to determine the individual clusters of grains and their respective centroids. Ice grains that are not assigned to one of the clusters (small black dots in Figs. 2a – 2g) are not considered for the velocity calculation as they mostly appear within fracture traces (particularly in 22 and 45 m). Further details about the crystal structure are discussed in Hellmann et al. (2020b).

We calculated the acoustic velocities from the COF patterns of all samples. The resulting velocity distributions (Figs. 2h – 2n) are functions of azimuthal direction and inclination (i.e. colatitude, 0° parallel to vertical core axis). The velocities were calculated on a dense grid for azimuth and inclination angles of the incident seismic wave with 1° for both angles to avoid interpolation artefacts. The direction of the maximum velocity in each sample coincides in general with the centroid of the multi-maxima cluster (Fig. 8a – Fig. 8g). The exact position of the centroid and thus the velocity maximum depends on the weighting factor that considers the size of the individual ice grains. This may lead to a slight offset between the geometrical midpoint of the diamond-shape pattern and this centroid of the multi-maxima pattern. The minimum velocity is found on a small circle with an opening angle of about 45° around the centroid. Perpendicular to the centroid, another minor velocity maximum along a girdle can be observed. The median value of ice velocity per sampling depth lies between 3834 and 3840 m s$^{-1}$. The anisotropy generally increases with depth (Table 1) and reaches a maximum value of

$$\frac{\max(v_p) - \min(v_p)}{\max(v_p)} = 2.32\,\% \tag{1}$$

at 79 m between the global maximum (around vertical direction) and minimum velocity values (Fig. 8n).

The p-wave velocity for vertically incident waves (parallel to z-axis of the core) increases with depth, especially for the deepest parts, where the cluster is centered around the vertical axis (Fig. 3a blue line). The p-wave velocities for a colatitude of $\varphi = 90°$ (horizontal direction) are shown in Fig. 3b (mean value per sample) and Fig. 4. The largest azimuthal variations appear at 2 m since the c-axes of the grains cluster around a horizontally oriented centroid ($\varphi_c = 88.6°$). The maximum horizontal anisotropy is 1.4 %.

### 3.2 Acoustic velocities from ultrasonic experiments

We measured the acoustic velocities on five of the above-mentioned ice core samples. The ice core samples were taken from 2, 22, 33, 45 and 65 m depth and usually from the upper of the two ice core segments that have been used for the COF analysis (cf. Fig. 1c). The distance between the uppermost COF thin section and the ultrasonic sample is between 5 and 15 cm (with an exception for 65 m with an offset of 60 cm). For each sample, we carried out three individual horizontal measurements





**Table 1.** Mean, minimum, maximum p-wave velocity and grade of anisotropy for each sample.

| depth [m] | 2 | 22 | 33 | 45 | 52 | 65 | 79 |
|---|---|---|---|---|---|---|---|
| $\bar{v}_p \, [\mathrm{m\,s^{-1}}]$ | 3824.5 | 3824.8 | 3825.7 | 3825.9 | 3824.1 | 3830.7 | 3837.9 |
| $\min(v_p) \, [\mathrm{m\,s^{-1}}]$ | 3808.3 | 3804.5 | 3802.2 | 3807.8 | 3800.9 | 3805.3 | 3798.7 |
| $\max(v_p) \, [\mathrm{m\,s^{-1}}]$ | 3862.8 | 3864.2 | 3881.1 | 3871.6 | 3863.4 | 3873.6 | 3889.0 |
| anisotropy [%] | 1.41 | 1.54 | 2.03 | 1.65 | 1.62 | 1.76 | 2.32 |

on three different levels (indicated as $z_1$, $z_2$, $z_3$ in Fig. 1a). For the vertical measurements, we obtained one measurement per sample. As there was a half cylinder of ice from the lowermost depth (79 m) available, we also measured the vertical velocity for this sample. The ultrasonic measurements were conducted using different pieces of ice than used for the COF analysis and therefore, the actual grain size and distribution remain unknown. The positions of the ultrasonic measurements are marked in

Fig. 2 by black dots.

In a first step, the recorded traces were shifted to correct for zero-time $t_0$ and the p-wave arrivals (example shown in Fig. 1b) were picked. Additionally, the ice core diameter for each azimuth was measured and since the core was not perfectly round, the diameter varied by a few millimeters. The velocities for each azimuth were calculated using the ice core diameter and the p-wave travel time.

To ensure data consistency, the reciprocal travel times were compared for quality checks. Rays with opposing azimuths ($\vartheta$ and $\vartheta$+180°) are reciprocal and the velocity should be identical. Larger deviations ($>30\,\mathrm{m\,s^{-1}}$) for individual measurements were considered incorrect and these measurements were removed from the final dataset (in total, 7 out of 315 traces). Finally, the reciprocal traces for the individual horizontal and vertical measurements were combined and an average velocity for each azimuth was calculated. That is, we only consider an azimuthal range of 0-180° and therefore, the horizontal results show a

periodicity of 180°. This processing scheme was applied to all 5 samples and the results are summarised in Fig. 4. Minimum and maximum velocity within the stack of repeated measurements for each azimuth are shown as reddish coloured areas.

All 5 samples show a set of 2 maxima surrounded by 4 minima and 2 local side-maxima. The positions of the maxima for measured and COF-derived profiles coincide within a range of a few degrees of azimuth. The measured velocity profiles show higher amplitudes between maximum and minimum compared to the calculated COF-derived profiles, but the latter are rather

smooth. The velocities (Fig. 3a) increase with depth, which is in accordance to the COF-derived profile.

### 3.3 Porosity from X-ray tomography

The X-ray CT images provide porosity information in the vicinity of the horizontal ultrasonic measurements (summarised in Table 2). The porosity is governed by air bubble layers in the ice. These air bubble layers show a preferentially horizontal distribution and alternate with air bubble free layers along the entire ice core borehole as shown by images of an optical televiewer

(OPTV, images not shown). We calculated individual values for each sample and the average porosity over all five samples (0.682%). An additional porosity analysis based on 2D Large Area-Scanning Macroscope (LASM) images (Binder et al., 2013;





Krischke et al., 2015), obtained during the thin section preparation, produced similar results (LASM-derived average poros-
ity 0.645%). In contrast to the porosities from three-dimensional CT measurements, the porosity values determined from the
two-dimensional LASM images continuously decrease with increasing depth. This indicates an increasingly heterogeneous
dirstribution of air bubbles in deeper parts of the ice, since the porosity values derived from the LASM images are averaged
values obtained on 50 cm of ice (Fig. 1c).

The individual CT-derived porosity values (Table 2) are taken into account for the COF-derived vertical velocity profiles in
Figs. 3 and 4 (blue curves). This correction reduced the COF-derived velocities by about $30\,\mathrm{m\,s^{-1}}$, when assuming air-filled
spherical bubbles as second phase (cf. Fig. 3a (dashed magenta line) as an example for uncorrected values). Since we have a
relatively low porosity (<1 %), but do not know the exact size and position of the individual air bubbles, we used a correction
for spherical inclusions at very low volume fraction, where the effective elastic moduli can be calculated exactly Torquato
(2002, p. 499). The CT and LASM images indicate that the majority of air bubbles not associated with grain boundaries are
spherical and do not show any elongation in certain directions and therefore confirm our assumption. We retrieved the required
bulk and shear moduli of the ice matrix from the corresponding elements of the computed polycrystal elasticity tensor. With
these bulk and shear moduli, the CT-derived porosity values and the mass densities of air ($\rho = 1.3163\,\mathrm{kg\,m^{-3}}$ at $T = -5°$)
and ice ($\rho = 918\,\mathrm{kg\,m^{-3}}$), we obtained the mean velocities of such a two-phase material. Finally, the difference between this
mean velocity and the calculated mean velocities of pure ice was subtracted from the individual velocity values. This correction
was applied to the COF-derived profiles (blue curves) in Figs. 4 and 3. The porosity correction causes a shift of the average
velocities (see Fig. 3 blue dashed vs. solid lines), but does not affect the shape (i.e. maxima and minima) of the horizontal
profiles at the individual depths (Fig. 4).

**Table 2.** Porosity values [%] for each ultrasonic sample (derived from CT measurements) and the corresponding COF thin sections (derived
from LASM scans with a vertical offset of 10-15 cm to CT-samples).

| depth [m] | 2 | 22 | 33 | 45 | 65 | mean |
|---|---|---|---|---|---|---|
| porosity (CT) [%] | 0.63 | 0.27 | 0.81 | 0.35 | 1.35 | 0.682 |
| porosity (LASM) [%] | 1.44 | 0.55 | 0.68 | 0.23 | 0.32 | 0.645 |


## 4    Discussion

### 4.1    Comparing COF-derived velocity and ultrasonic measurements

The results for COF-derived velocities and the ultrasonic velocity profiles are compared in Fig. 3 and Fig. 4. As presented in
Sect. 3.3, the COF-derived velocity profiles were corrected for the porosity. The vertical velocities (i.e., parallel to the ice core
axis), shown in Fig. 3a, show a relatively good match between the two methods. Likewise, the average horizontal velocity pro-
files (Fig. 3b) coincide well within the uncertainty ranges. This uncertainty range is defined by the standard deviation around
the mean value for all azimuths.





Azimuthal variations of the horizontal measurements are compared in Fig. 4. Only the sample from 22 m depths shows reasonable matching with respect to the positions and amplitudes of the velocity minima and maxima. For all other depths, there are

considerably large differences between COF-derived and ultrasonic velocities. It is noteworthy that the sample at 22 m depth exhibits a lower porosity (i.e. lower amount of air bubbles) compared with the remaining samples (Table 2), but the air bubble content cannot fully explain the observed discrepancy between the two velocity profiles shown in Fig. 4. These discrepancies could be caused by the differences in the grain size distribution within the individual samples, since we did not conduct both measurements on exactly the same pieces of ice.

Seismic waves have a band-limited frequency content resulting in a finite range of wavelengths. As indicated in Sect. 2.3, the dominant wavelength for the ultrasonic measurements was approximately 3.8 mm. As a consequence, the seismic waves are not just affected by the medium along an infinitely thin ray path connecting the source and receiver, but by a finite volume surrounding the ray path. This volume can be estimated with the first Fresnel volume path (e.g. Williamson and Worthington, 1993). Assuming a homogeneous medium including source position S and receiver position R, a point D is considered to be

within the first Fresnel volume, when

$$\overline{SD} + \overline{DR} - l \leq n\frac{\lambda}{2}, \tag{2}$$

where $l$ is the direct ray path between source and receiver, $n$ is the order of the Fresnel zone, and $\lambda$ is the dominant wavelength. The ice grains within this Fresnel volume influence the velocity that is derived from the corresponding ultrasonic measurement. To illustrate the situation, we superimposed in Fig. 5 a Fresnel zone computed from Eq. (2) on one of our thin section

of temperate ice including large grains. Figure 5 shows that not only the size but also the position of the particular grains may influence, how significantly grains of the particular clusters affect the final velocity profile for the ultrasonic experiments. If grains of a certain cluster only appear at the margins of the ice volume (e.g. the dark blue grains) only a few measurements are affected by these grains and thus the overall effect of this cluster is smaller compared to its actual statistical appearance in the ice core volume. The analysis becomes even more complex when considering the shape of the grains. As observed in our core

data and also in earlier studies (e.g. Hooke and Hudleston, 1980; Monz et al., 2020) the grains in temperate ice are branched. In order to get a statistically significant COF pattern, the COF analysis considers a relatively large number of grains, which leads to an averaged associated velocity profile. In contrast, a few branched large grains may be quite prominent in the actual ice volume (see Fig. 5) selected for the ultrasonic measurement. Thus, these few large grains are dominating the measured velocity profile.

Ultrasonic measurements analyse a limited volume in the vicinity of a particular source-receiver pair, whereas the COF analysis with a higher number of grains within a larger section area provide integrated velocities over larger parts of the ice core. Therefore, the velocities of COF analysis and the ultrasonic measurements are expected to be different in the presence of large grains. Conversely, a good match can be expected, when a large number of small grains is involved.

To investigate this further, we computed grain size distributions (Fig. 6) using all thin sections prepared for the COF analysis

(Hellmann et al., 2020b). Clearly, the sample from 22 m depth shows the largest number of grains and thus the smallest mean grain size. Considering the previous discussion, it is therefore not surprising that we observe a relatively good match in Fig. 4





for this sample and larger discrepancies for the remaining samples.

As a result of the previous discussion, we also assume that a larger amount of ultrasonic measurement levels should lead to a better match with the statistically averaged profile from the COF analysis. Additional ultrasonic measurements are avail-
able for the sample of 33 m. These measurements were obtained on the neighbouring core segment just below the analysed COF samples (the original ultrasonic measurements, shown again in Fig. 7a, were acquired above the COF samples). When considering the additional measurements the differences between the mean velocity profiles derived from COF analyses and ultrasonic measurements further decrease (Fig. 7b). In turn, when considering only a subset of thin sections (here only the four horizontal thin sections) to derive the velocity profile, it also converges to the ultrasonic profile (Fig. 7c).

To conclude, ultrasonic and COF analyses complement each other. The first is a deterministic approach allowing a detailed analysis of a particular ice core volume. The latter is a statistical approach that provides an integrated COF pattern derived from several centimetres (up to 50 cm) long ice core samples and thus an averaged velocity. However, both methods are most likely comparable, when the numbers of grains are similar in both samples. Hence, both methods should be combined and ultrasonic measurements may become a valuable technique to support the existing method.

## 310  4.2   Ambiguities with other COF patterns

In this study, the COF patterns are assumed to be known a priori and the ultrasonic results could be correlated with this known COF. The question arises, if ultrasonic measurements are a suitable method to determine unambiguously unknown COF patterns?

To address this question, we consider the sample at 22 m depth. Its resulting COF and the associated velocity distribution,
already shown in Figs. 2b and 2i, are shown again in Figs. 8a and 8b. For this sample, the small grain size prerequisite is met, leading to a good match between COF-derived and ultrasonic velocity profiles (Fig. 4b, shown again in Fig. 8e without uncertainty ranges). We now compare the results with a small circle girdle structure (Fig. 8c), which is a common COF pattern for compressional deformation in combination with recrystallisation (Wallbrecher, 1986). Its corresponding velocity distribution is shown in Fig. 8d. When only considering the horizontal orientations, as measured with our ultrasonic experimental setup
(marked with black dots in Figs. 8b and 8d), the azimuthal velocity variations of both, the actual "diamond shape" pattern (blue curve in Fig. 8e) and the small circle girdle (dashed green curve in Fig. 8e), are compatible with the measured ultrasonic data (orange curve in Fig. 8e). Obviously, there exist several COF structures that explain the ultrasonic data equally well, thereby leading to ambiguities and uncertainties in interpreting the ultrasonic data. Similar ambiguities can be observed for COF patterns typical for polar ice such as single maximum vs. girdle fabrics. Adding the additional vertical measurement parallel to
the core axis (black dot in the centre of Fig.8b and 8d) does not remove this ambiguity for the small-girdle example.

To further reduce this ambiguity, it would be required to add additional ultrasonic measurements, spanning a range of azimuths and inclinations, such that the area of the stereoplots would be sampled more regularly. With modern point-contact transducers, it seems to be feasible to implement such an experimental layout with reasonable expenditure of time when using a multi-channels recording system.

These ambiguities show, that COF analyses will also be required in the future, but ultrasonic measurements can support this



analysis and bridge the gaps between the discrete COF samples. Finally, ultrasonic measurements on ice cores and in borehole provide the link between COF and surface geophysical velocities (Bentley, 1972; Gusmeroli et al., 2012; Diez et al., 2014).

## 5 Future technical improvements

Our measurement scheme (Fig. 1a) was built for first attempts to investigate the feasibility of ultrasonic measurements to detect
the COF along an ice core and to establish a link between COF and cross-borehole or surface seismic experiments. Although we showed that there exist ambiguities, such a device provides valuable information and could directly be employed in-situ on freshly drilled ice cores. As an advantage of an immediate measurement on thermally drilled ice cores, one would avoid the refreezing of meltwater and thus a much better coupling of the transducers without extra work for removing this meltwater "skin". For mechanically drilled cores with a relatively convex shape, the point contact transducers are expected to be well
coupled. Furthermore, more than two transducers are recommended to obtain several inclined measurements as discussed above and the transducers should further be pressed onto the ice with a defined constant pressure. A constant pressure is relevant to avoid any pressure melting effects and ensures identical coupling conditions. This enhances the comparability of the acoustic signals throughout the entire experiment.

In addition, the determination of the exact distance between source and receiver should be automated. A manual measurement
of the distances, as performed in our experiments, leads to a higher uncertainty in the derived velocities. Moreover it is not feasible with several transducers. These improvements require a more comprehensive measurement device. Such a device could be employed in a processing line (e.g. in polar ice core drilling projects) with existing devices such as for Dielectric Profiling (DEP) (Wilhelms et al., 1998) before cutting the ice core into sub-samples for different analyses. As it also allows for a fast data acquisition, such a device could also be employed for other purposes such as detecting the link between two neighbouring
ice core segments (i.e. retrieving the actual orientation of the freshly drilled segment within the glacier).

## 6 Conclusions

We have performed ultrasonic experiments at ice cores from a temperate glacier, and we compared the results with those from a well-established COF analysis method. The main objectives of this study were (i) to compare the ultrasonic and COF-derived seismic velocities and (ii) to check, if ultrasonic measurements have the potential to replace or reduce the labour-intensive and
destructive COF analysis. Our main findings can be summarised as follows.

- Ultrasonic and COF-derived seismic velocities are comparable, when the grain size of the ice crystals is sufficiently small. However, this condition is generally not met in temperate ice. In contrast, we recommend to apply this method to cold (e.g. polar) ice cores with small grains.

- In the presence of large grains, we observe a poor correlation between the ultrasonic and COF- derived velocities. The
ultrasonic measurements belong to the deterministic approaches. Each measurement samples the actual 3D volumes (Fresnel volumes) and only considers the grains therein. The COF-derived profiles provide a statistical mean value of



the velocities for all thin sections. Therefore, the number of measurement levels of ultrasonic measurements needs to be sufficiently large. This is especially relevant for samples from temperate ice cores.

– In the presence of a significant porosity (i.e. air bubbles), a correction needs to be applied, to make ultrasonic and COF-derived velocities comparable. This requires the determination of the porosity. In this study, we have employed a CT scanner for that purpose.

– In principle, ultrasonic measurements can be employed for determining COF patterns. However, this requires a relatively dense sampling of the ice core, including a broad range of azimuths and inclination angles. Our experimental setup, including only horizontal and vertical measurements, led to ambiguous results.

On the basis of our findings, we conclude that ultrasonic measurements are not yet an adequate replacement of COF analysis. However, since the development of ultrasonic transducers is progressing rapidly, we judge it feasible that adequate experimental layouts of ultrasonic experiments can be implemented in a foreseeable future. This would offer substantial benefits, since it would reduce the labour-intensive COF analysis. Furthermore, the ultrasonic measurements offer the significant advantage of being non-destructive, and the samples of the generally valuable ice cores would remain available for other analyses of 375 physical properties. This also means, that the ultrasonic measurements can continuously be obtained on freshly drilled cores. Nevertheless, a certain but reduced number of thin sections a COF analysis can still be used to calibrate the ultrasonic data and to dispose of ambiguities with direct comparisons of the results of both methods on the same ice core samples.

*Data availability.* The ice fabric data and the LASM images are published in the open-access database PANGAEA® (Hellmann et al., 2018a,b). The ultrasonic data are available in the open-access database ETH Research Collection (Hellmann et al., 2020a).

*Author contributions.* This study was initiated and supervised by HM, AB and IW. SH, JK and IW analysed the ice core microstructure to obtain the COF and calculate the seismic velocities. SH, MG and HL planned and conducted the ultrasonic and CT measurements. Data processing and calculations were made by SH with support from all co-authors. The paper was written by SH, with comments and suggestions for improvements from all co-authors.

*Competing interests.* The authors declare that there are no conflicting interests.

*Acknowledgements.* This project is funded by the Swiss National Science Foundation under the SNF Grants 200021_169329/1 and 200021_169329/2. Data acquisition has been provided by the Paul-Scherrer Institute, Villingen, the Alfred Wegener Institute Helmholtz Centre for Polar and Marine Research, Bremerhaven and WSL Institute for Snow and Avalanche Research SLF, Davos. We especially thank M. Jaggi, P. Sel-





vadurai and C. Madonna for their extensive technical and scientific support for the ultrasonic measurements and the provided equipment and T. Jenk, M. Schwikowski and J. Eichler for their support during ice core drilling and processing.



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





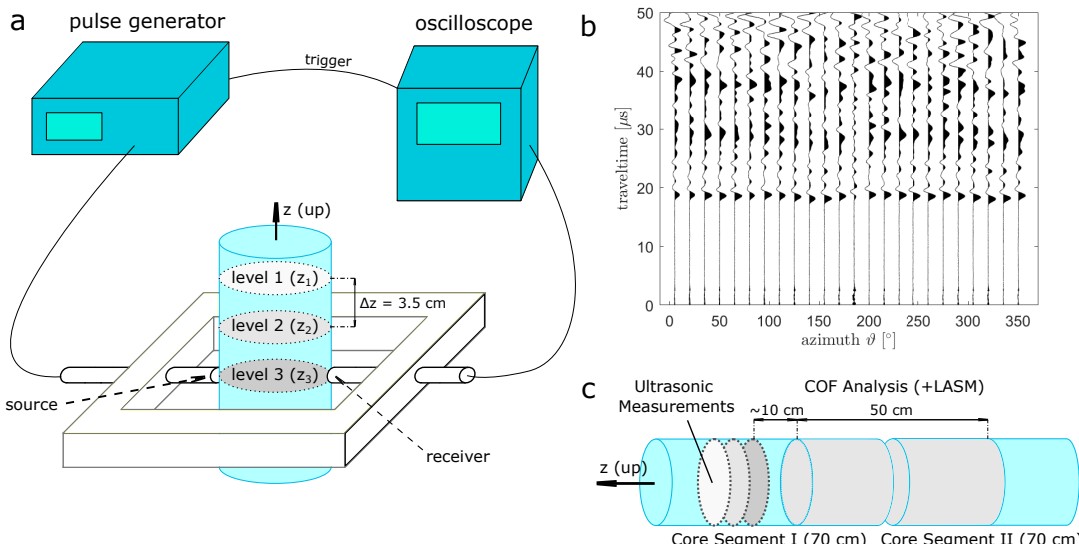

**Figure 1.** (a) Schematic experimental setup for ultrasonic measurements on the ice core with tools and devices (amplifiers not shown). (b) Example dataset of seismic traces for horizontal measurements. (c) The segmentation of two adjacent ice core segments depicting the position of the ultrasonic and COF analyses.





**Figure 2.** COF patterns and calculated seismic velocities for all seven analysed ice core samples: (a-g) c-axis distribution on a lower hemisphere Schmidt plot (the core's vertical axis aligns with the centre of the plot). The grains associated with the individual clusters are colour-coded respectively. (h-n) Seismic p-wave velocities (derived from COF) for any azimuthal direction and incident angle plotted on a lower hemisphere net. They were plotted with the MATLAB Toolbox MTEX (Mainprice et al., 2011). The black dots symbolise the sets of angles ($\vartheta/\varphi$) for the ultrasonic measurements.





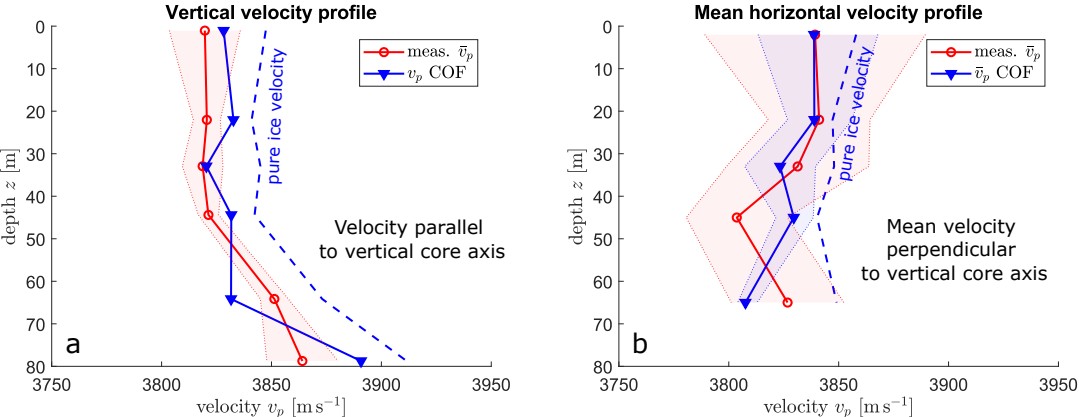

**Figure 3.** Mean values for measured and calculated seismic velocities for (a) vertical, i.e. $\varphi = 0\,^\circ$ and (b) horizontal direction, i.e. $\varphi = 90\,^\circ$ along the ice core. The shaded areas show the standard deviations of the respective measurements. The dashed blue lines show the respective COF-derived velocities without porosity correction.



**Figure 4.** Mean measured seismic velocities from ultrasonic experiments (orange curve and red dots) with maximum and minimum values (light red areas) for five ultrasonic samples and the corresponding calculated velocities from the COF distribution (blue curve) from Fig. 2 with Voigt and Reuss bounds. These datasets are for horizontal measurements, $\varphi = 90°$. Depth indicated in upper right corners.



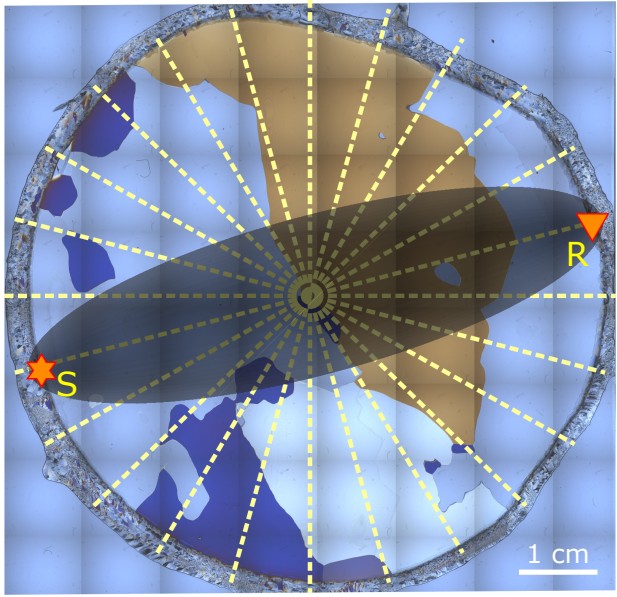

**Figure 5.** Raw image from fabric analyser, showing a typical grain distribution found in the temperate ice core. The coverage by ultrasonic measurements (dashed lines) and an example for the first Fresnel zone (homogeneous medium approximation) are superimposed, S is the sending transducer and R is the receiver. The distance between source and receivers is $\approx 7\,\mathrm{cm}$ on average.





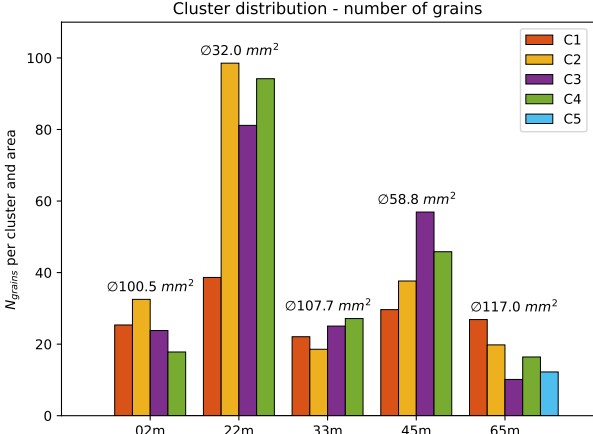

**Figure 6.** Number of grains over all clusters in the individual samples. The mean grain size per sample is noted above. The clusters are colour-coded according to Fig. 2.





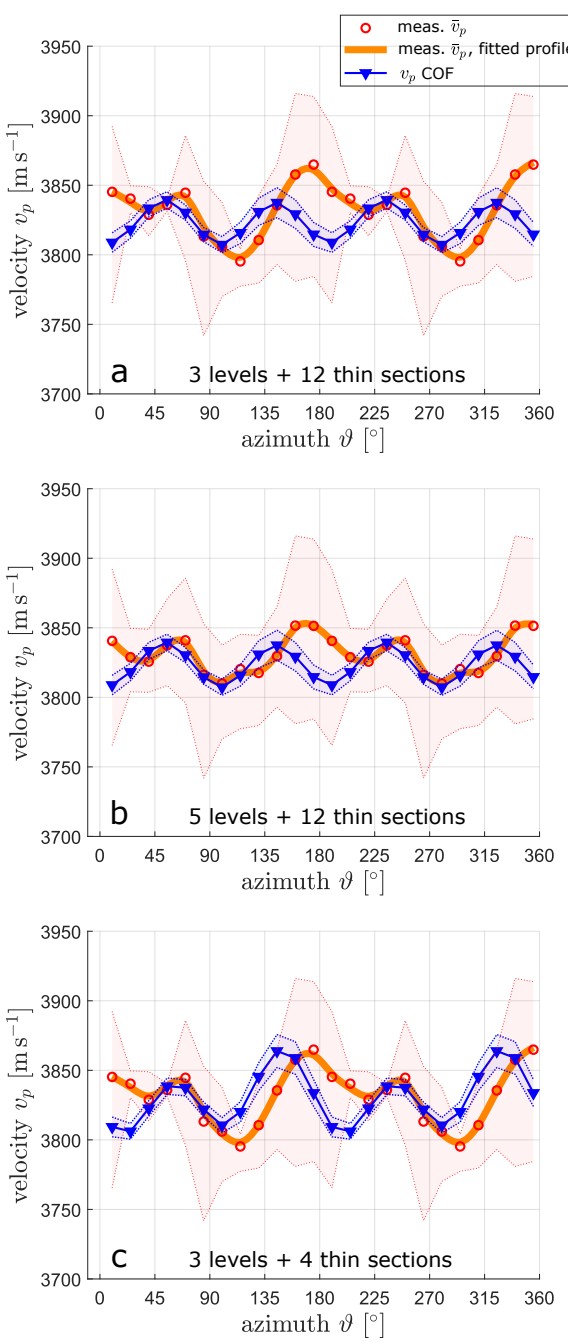

**Figure 7.** Mean measured seismic velocities from ultrasonic experiments (orange curve and red dots) and the corresponding calculated velocities from the COF distribution (blue curve) for the sample in 33 m, (a) same as Fig. 4c, (b) with a larger amount of ultrasonic measurements, (c) with a smaller amount of thin sections considered for the calculated COF-derived velocity profile.



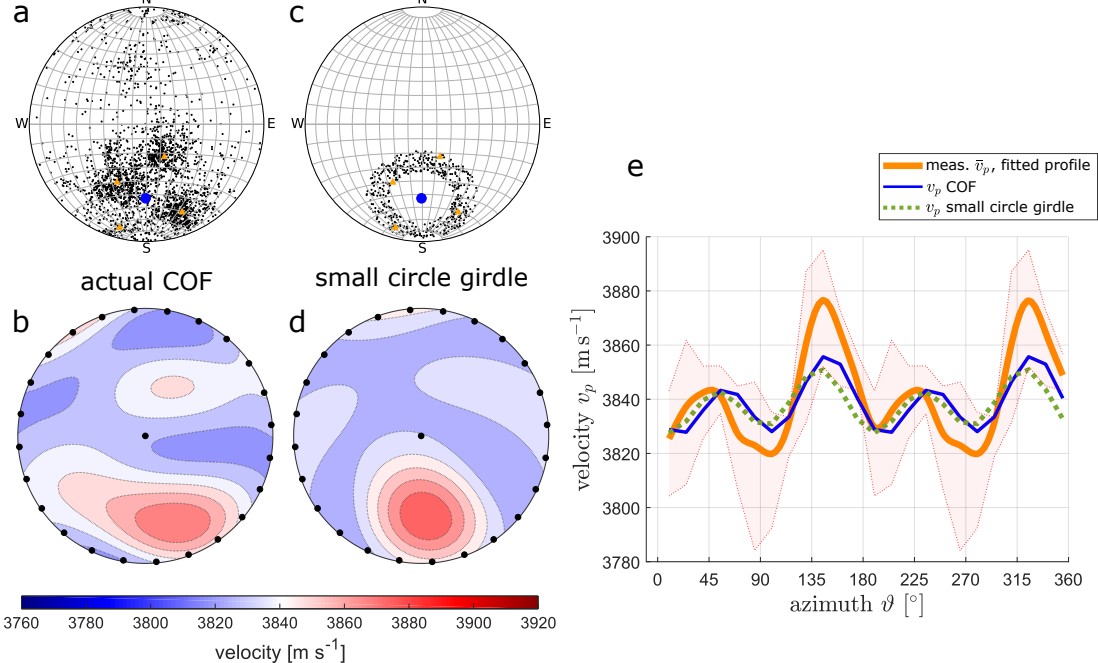

**Figure 8.** Comparison of different fabric types and their velocity profiles for different inclination angles: (a) Lower hemisphere Schmidt plots for actual multi-maxima fabric for the sample from 22 m; b) the derived acoustic velocity pattern for (a). (c) Lower hemisphere Schmidt plots for a modelled small circle girdle fabric; (d) the derived acoustic velocity pattern for (c). The black dots in (b) and (d) symbolise the horizontal and vertical measurement positions. (e) The velocity profiles for horizontal measurements: orange line - actually measured profile, blue line - velocity profile of actual COF, dashed green line - potential velocity profile for a small circle girdle.