# Peer review of "Acoustic velocity measurements for detecting the crystal orientation fabrics of a temperate ice core"

_The Cryosphere, 2021_

## Author Comment (AC1)

Kia ora Dave,

Thank you very much for your recommendations to our paper about acoustic velocity measurements. In the following, we include your comments in italic followed by our answers in normal font.

*Hi Sebastian. Interesting paper. I have a few comments - sorry read the paper about 6 weeks ago and then ran out of time to post comments. These done in a rush to meet discussion deadline - so hope they make sense.*

*Your lab measurements in fig 4 have a sinusoidal variation with a peak to peak separation of about 90 degrees. This pattern is there at 22 and 33m in the COF calculated Vp pattern, but not really at 2, 45 and 65m. The measurements at and calculations at 33m appear to be phase shifted by ~ 30 degrees. This makes me wonder whether your core cross section is a circle or something else. An eccentric shape (circle flattened in orthogonal directions towards a square shape) would give this pattern. 1mm of eccentricity would give ~74m/s variation in velocity if diameter were always assumed to be the same (at 68mm).*

*I'm not sure what would cause an eccentric shape- an elliptical shape would be easier to understand (relaxation?)- whatever some clarity as to whether your cross sections are actually circular would be important. In our work on mechanically recovered cores we have found that the diameter is not constant for different azimuths at a single depth. I do not have enough data to say what the pattern is (if there is a consistent pattern) but we have two solutions. One student measured the diameter corresponding to each particular azimuth individually. This is very time intensive and I don't recommend it. In more recent work we have machined the samples on a lathe to get a diameter that is constant (with azimuth at a single depth) with a tolerance of better than 0.1mm. This approach is straightforward and effective.*

*The text around line 160 does not make it clear whether you have measured the diameter for each azimuth or whether you are assuming that diameter is constant with azimuth at a particular depth. I think you need to clarify here. You quote a statistic of 68+_0.36mm. If you take this at face value then there will be a velocity error of +_ 26m/s on each measurement. You should show this as an error bar on fig 4.*

*I have other comments scribbled on the manuscript and can send that to you if it is useful- just email me.*

We agree that the core diameter has a significant influence on the particular velocity for each azimuth. Similar to your first student's work, we decided to measure the core diameter for each single measurement individually. This was time consuming. If future measurements will be carried out along polar ice cores, some automated techniques should be considered. We also lathed the core, since our core samples were covered with a very irregular meltwater "skin". This skin has formed due to the thermal drilling technique and was frozen to the core when the segments were stored at -30°C.

We appreciate your comment and have clarified in our revised version that we measured the diameter individually for each measurement and that these core segments were lathed.

The diameter given in line 160 is only the average derived from all the individual measurements to provide a diameter range.

Best,

Sebastian and the co-authors

---

## Author Comment (AC2)

Dear Sridhar Anandakrishnan,

Thank you very much for your very motivating and positive comment. In the following, we include your comments in italic followed by our answers in normal font.

*This is an excellent paper ready for publication. The results are new and important. The writing is clear and elegant. The figures are well-drafted and complete.*

*When working with ice core crystal orientation fabric, there are three paths: the crystallographic/mineralogic where you examine thin sections under polarized light; the geophysical, where you study the speed of seismic waves at various angles; and the numerical, where you estimate the aggregate behavior of a collection of crystals from the behavior of a each single crystal combined in the aggregate. Most researchers focus on one or the other; these authors have tackled all three successfully. They have combined the classic and never-improved-upon work of Bennett (1968) with modern methods of estimating aggreagate behavior. They have gone one better, by including the effect of voids by analyzing the bulk structure using x-ray imaging.*

*The seismic profiling was conducted at 1MHz (wavelength smaller than individual crystals) and found good agreement between their estimates from the model and the measurements. They note that while one can validate COF measurements from the ultrasound data, it is difficult to use the ultrasound data to infer COF - the density of raypaths needed is impractical. I note that previous work (Bennett, in particular) used a much simplified model for aggregation (cones of various sizes and directions). A comparison of the error from the Bennett method and this more-complete method would be illuminating. In particular, if the Bennett method isn't grossly inaccurate, then fewer ultrasonic raypaths would be needed to solve for the fabric. That could be a starting point to an improved, more-correct solution.*

*2nd, the scaling from 1MHz to 100Hz (mm wavelength vs 10s of m) should be briefly addressed - what level of detail is needed in the model when we are collecting data using explosive or vibroseis seismic data in the field? There is no point in me belaboring the point - I liked it. People should just read the paper.*

In our paper, we could only investigate two of the three fields. The geophysical application towards surface or borehole seismic applications is still in progress. The modelling approach does not consider the frequency of the seismic waves and is a more general model to calculate seismic velocities from a given elasticity tensor.

In our outlook, we propose that ultrasonics can fill the gap between COF-measurement and surface/borehole seismics but at the current stage, we only have some first ideas how to upscale these measurement results to surface/borehole seismic measurements. One of the remaining issues is the frequency shift (MHz->kHz) that plays an important role when comparing ultrasonic and in-situ seismic data. We hope, that some additional investigations can contribute in the future.

Kind regards,

Sebastian Hellmann
on behalf of the Co-Authors

---

## Author Comment (AC3)

Dear Valerie Maupin,

Thank you very much for your valuable comments and suggestions. In the following, we include your comments in italic followed by our answers in normal font.

*The article compares two methodologies to measure the seismic anisotropy in an ice core. Considering the relation between ice dynamics and anisotropy, and the fact that seismic investigations provide a non-invasive methodology for mapping glacier properties, this is of course an interesting and important contribution. It shows also intriging results.*

*The article is very well written. The methodology is very well described and the results are well presented. The sonic records shown in Figure 1 are very nice and should ensure very good quality data. My only major comment concerns the interpretation of the difference between the results of the two methods. I am not completely convinced by the explanation that the limited number of grains in the ultrasonic case is the reason for the difference, and, from the figures, I think that the difference is larger than the text gives an impression of.*

*Figure 4 is central to compare the results of the two methods. The ultrasonic tests show a rather wide band of measurements, which is absolutely normal, and they have a clear trend, but I notice that the COF do not even fall within this band for some azimuths, at 2 and 65m depth. The authors argue that the discrepancy between the two methodologies come from a less representative sampling of the ultrasonic measurements. In favor of this, I notice that the ultrasonic measurements show a higher amplitude of anisotropy, which would fit with the fact that they represent one orientation, rather than the averaging done by COF, but they also claim that Figure 7, where more measurements were done, confirm their hypothesis. I agree that the amplitudes match better in the b) and c) plots than in the original a) plot, but the dominant shapes and positions of the maxima of the blue and red curves do not change from plot to plot. What rather strikes me is actually the consistency of the red curves between the three plots in Figure 7. That would suggest to me that the number of grains is not the main factor creating the difference, and that the difference is a systematic difference in how the two methodologies view the anisotropy. The authors show the dimension of the Fresnel zone in Figure 5. The Fresnel zone is actually a volume that also extends in the vertical direction. Waves propagating in this volume propagate in slightly different directions. I would therefore assume that the velocity seen by the ultrasonic tests is not exactly the one calculated in the source-station direction, but an average around this direction. I notice in Figure 2 that maximum velocities (considering all dips) often occur in an azimuth not very different from the azimuth of the maximum of the ultrasonic measurements. In particular at 45m depth, the max is at about 135degrees, in the same azimuth as the max velocity for the ultrasonic measurements. Of course the dip with respect to the horizontal plane is not small. The anisotropy is rather small here. I do not expect this would distort the shape of the Fresnel zone or give a very different group and phase velocity direction. I do not claim contributions from off-plane directions is a good explanation, but I think it would be worth exploring it a bit in the text, as an alternative.*

We have considered your point on off-plane reflections. Indeed, this is an important point that required clarification. However, the ultrasonic measurements do not really suffer from off-plane effects. Other than the COF-analysis, the ultrasonic experiments allow a full three-dimensional measurement of a particular amount of ice. They are more suitable to consider the shape of the branched, large grains. We already tried to discuss this in our discussion section, but did not state this so explicitly. In our revised version, we have added that the Fresnel Zone is a volume and that in particularly the very branched and interlocked grains in the temperate ice would lead to a significant difference between the velocity derived from 300 µm thin ice core samples and a $cm^3$-large ultrasonic volume.

However, even though the Fresnel Zone is a volume of a few $cm^3$, the total number of grains is rather limited. We are still convinced that the large grains and the associated clustering effect due to recrystallization (SIBM-N) are the major driver for the differences.

Nevertheless, we also agree that the COF-derived profiles are lacking some information from out-of-plane effects, but this is again an issue driven by grain size and grain clustering. In polar cores with

grains of a few mm, we expect that the effect is much less important when assuming a statistically uniform distribution of the different clusters in the samples. We plan to test this hypothesis on polar cores in future projects.

In conjunction with your comment for L288, we also added some further details about the "parent grain" pattern observed in the ice core. Large grains with a certain orientation are often surrounded by many smaller grains with a similar orientation. This is an observation within the context of recrystallization mechanisms in the temperate ice → strain-induced boundary migration with nucleation of new grains (SIBM-N, see our first paper Hellmann et al., 2021).

Here are our answers to your line-specific comments:

*Figure 4 is a very central figure. The data are actually duplicated from a 0-180 to a 0-360 degrees range. I think this might increase artificially the impression of fit and should be avoided. It would be interesting to have the vertical velocity in the same figure, as an extra small column to the right for example, in order to exploit more the vertical direction velocity in the interpretation.*

We adjusted the Figure (0-180° range) and added the vertical measurements to Fig. 4 (and Fig. 6). For clarifying that these measurements contain a periodicity, we have added the first/last measurement to the end/beginning of the profile.

*line 14: "concise": should be "consistent"?*

Indeed, this should be consistent, changed.

*Figure 2 is cited before Figure 1, as far I can see, and you should normally exchange the figure numbers. As Figure 1 is a good overall summary of your set-up, find a way to cite it before?*

We added a reference to Fig 1c before referencing this Fig 2a-g here.

*line 44-45: rephrase. "since..." does not really make sense with beginning of sentence.*

We rephrased this sentence as follows:
These methods investigate the elastic parameters of the ice. Since elastic parameters and COF are directly related, the methods can also be employed for COF analyses.

*line 59: benchmark to what?*

We replaced this term by "relevant measurement parameter".

*line 101: move sentence to line 128, as this gives the impression you won't give any details, but you give them afterwards, and they are necessary.*

We agree that this sentence may confuse the reader. Therefore, we combined parts of this sentence with text from line 128 and formulated a new sentence in line 129: "The calculations for the

polycrystalline tensor and acoustic velocities are described in more detail in Maurel et al. (2015) and Kerch et al. (2018)."

*lines 129-130: I do not understand what you are saying here. Your step 4 is a Voigt average (linear average of elastic tensors); when you say here "seismic velocities", do you mean you take the Voigt average (and Reuss and Hill) to calculate the isotropic mean velocities?*

The seismic velocities derived from the elasticity tensor vary around a mean value, called Voigt bound, and those seismic velocities calculated from the compliance (stiffness) tensor provide variations around the lower Reuss bound. These two ways of calculating seismic velocities from the two reciprocal tensors provide an upper and lower bound. Both ways are possible and we use their results to get a velocity range for our calculated values. The mean isotropic values can be regarded as baseline for each calculation. The anisotropic values vary around this baseline.

We rephrased this sentence as follows:
The seismic velocities can be calculated from the elasticity tensor or the inverse compliance tensor. Both approaches provide velocity profiles oscillating around an upper (Voigt bound) and lower (Reuss bound) mean velocity (Hill, 1952). We calculated the seismic velocities from both tensors to obtain these upper and lower bounds of the potential velocity range and further derived the velocity profile from the Hill tensor (the mean of elasticity and compliance tensor).

*line 133: at this point you have not said you measure at -5deg. You have said you have frozen the core to -30deg.*

This is true. We rephrased the sentence and added in parentheses that the measured temperature is -5°C as described below. However, at this stage the exact temperatures are not necessary and in addition, they vary between ultrasonic and in-situ experiments.

*line 163: unclear sentence. The small wavelength does not favour that the individual measurements are a good integrated representation of the whole sample. Do you want to point out here that the wavelength is smaller than the grain size? Anyway, it is only the Fresnel zone dimension that matters to see if the wave field sees one grain at a time along its propagation, not the wavelength.*

We agree here, although we also want to point out that the size of the Fresnel Zone also depends on the wavelength (e.g. Lüth S., Buske S., Giese R. and Goertz A. 2005. Fresnel volume migration of multicomponent data. *Geophysics* 70, S121–S129).
However, in this part of the paper, we wanted to point out that we need to have a reasonably small wavelength to avoid effects as a result of wavelength vs sample size, e.g. if the wavelength is too large, stationary waves or waves that cause vibrations within the ice samples may be induced by the transducers. Those waves do not contain information about the COF. Furthermore, as we want to compare these ultrasonic measurements with in-situ data at a later stage, we also wanted to have a similar order of magnitude for the ratio core size – wavelength as in the in-situ seismic data (wavelength of around 3.8 m vs 100 m glacier thickness → factor of ~20, too).

We rephrased the sentence for clarification: "Thus, the wavelength is small enough to measure an integrated seismic velocity. This velocity can be regarded as the integrated velocity of the individual grain velocities. Much larger wavelengths may introduce geometric issues such as stationary waves, which are not representative for acoustic waves travelling through the glacier and thus would later inhibit a comparison with in-situ data."

*line 186: it seems there are many dark points within the clusters. It is not clear to me why they have been removed.*

Indeed, we excluded all grains below 0.5 mm$^2$ (< 1250 pixels). Those grains usually occur in fissures and as patches within the ice core. However, we reviewed the effect of these small grains and realised that they only minimally affect the calculated velocities (because the grain size is used as weighting factor for the sum over the velocities of the individual grains), visible changes only appear at 22 m and 45 m depth. Therefore, we also include these grains to avoid confusion. We changed Fig 2, 3, 4 and 6 respectively.
During this review, we realised that we used a wrong input file for the velocity calculations in 33 m and corrected this issue.

*line 199: would be good here to have the pure ice value for comparison.*

The given values in this section (incl. Table 1, Fig. 2) are the pure ice values. We clarified this in the captions of Fig 2 and Table 1. We applied the air correction for the first time, when comparing the data with the ultrasonic measurements (Fig. 3).

*line 218: indicate which uncertainty you have on this diameter, and if variation in diameter correlates in any way with the anisotropy.*

The diameter was determined manually for each individual measurement. In addition, the Figure below indicates that there is no correlation between the calculated velocity and the measured ice core diameter.

[Figure]

*line 228: The coincidence is not as good as stated by this sentence. The maxima for the COF and measured coincide only at depths 2 and 22m. For the three other depths, they do not coincide at all. At 45m, the maximum for the measured coincides with a minimum in COF.*

Together with the new Fig. 4, this discrepancy becomes more obvious and we extended this sentence as follows (to correct for this imprecision):
"All 5 samples show a set of 2 maxima surrounded by 4 minima and 2 local side-maxima. For the samples at 2, 22 and 65 m the positions of the maxima for measured and COF-derived profiles coincide within a range of a few degrees of azimuth (≤15°, Figs. 4a, b, e). At 33 m, there exist a significantly larger azimuthal shift (30°, Fig. 4c) and for the sample at 45 m maxima of one profile coincide with a minimum of the other (Fig. 4d)."

*line 230: One curve does not look like a smooth version of the other; I do not think you can blame the smoothness for the difference in amplitude.*

We rephrased the respective sentence: "The COF-derived profiles are in general rather levelled with smaller differences between the minima and maxima."
We do not want to say that one profile is a smoothed version of the other, but rather state that the COF-derived profiles are in general levelled profiles with smaller variations between max. and min. This observation is important for our discussion.

*line 230: This section is about the horizontal velocities, that do not increase with depth. You might remove this sentence.*

We have removed this sentence.

*line 247: you say that the air bubbles not associated with grain boundaries are spherical, but what about the grain boundary bubbles?*

Indeed, small grain boundary bubbles seem to be elongated as observed in many ice cores before. However, the larger bubbles, which are more frequent, seem to be hardly influenced by the grain boundaries. They are transecting and still appear round (see typical example in Hellmann et al., 2021, their Fig. 10). The reason is most probably the high temperature, that enhances the dynamics of the cycle of sublimation on large radius segments versus condensation on small radius segments, which takes place inside all bubbles and keeps them round as good as the dynamics allow.
We have added a sub-sentence in parentheses to clarify that bubbles on grain boundaries are influenced by the grain boundary processes and that other processes complicate an interpretation:
"(and therefore not pinned to and affected by the boundary pathways)".

*line 288: Is figure 5 representative of the number of grains? The mean grain size in Figure 6 does not really fit with the fact that a section of an ice core would have just a few grains. Is it such that grains in a given orientation cluster also tend to cluster in space? That would strengthen your theory if adjacent grains have the same orientation. Would it be relevant to calculate the velocity from the COF of one cluster only and compare with ultrasonic?*

The example in Fig. 5 is representative. The large number of grains arise from combining 9-12 of those thin sections (4 horizontal, 8 vertical cuts). In some of those thin sections, fracture traces with many small grains (especially at 22 and 45 m) were observed. Such fractures and patches of smaller grains further increase the total number of grains.

Due to the existing horizontal and two perpendicular vertical sections in south-north and east-west orientation (in total 3 perpendicular thin section from adjacent ice core samples, see Hellmann et al., 2021, their Fig. 4), we can actually see that the large grains are surrounded by smaller grains with a very similar orientation. Furthermore, the large grains are interlocked and branched and the ultrasonic measurement might be more sensitive to detect such an irregularly formed grain than a state-of-the art fabric analysis.

We also analysed the contribution of individual clusters and found that the measured velocity profiles can be explained with a combination of some of the four clusters. Usually, we observe that two clusters dominate, and the others seem to contribute less to the measured velocity compared to their occurrence (i.e. the relative grain area) in the thin sections. However, this is a rather qualitative investigation (no 1:1 comparison as the COF and ultrasonic experiments were not obtained on exactly the same ice). We do not have a profound physical explanation and assume that this is rather a statistical effect (several combinations of the clusters lead to similarly well-fitting results). Nevertheless, we added a sentence about this investigation.

---

## Author Response (AR1)

**List of changes according to the reviewer's comments:**

Substantial changes according to Ms. Maupins review comments:

1. *about potential off-plane reflections that may affect the comparison between ultrasonic and COF-derived velocities – We revised the respective part of the Discussion section (lines 298-318) and further discussed the influence of off-plane reflections and of the grains size:*

"Furthermore, we also observed a clustering of grains with similar orientation. In particular, small grains surround a larger grain, usually called parent grain, as a result of strain-induced grain boundary migration with nucleation of new grains (called SIBM-N, see Faria et al. (2014)). The irregular shape of the grains and the clustering of grains with similar orientation may lead to differences between the two velocity profiles. The Fresnel zone is actually a volume (third dimension not shown in Fig. 5) with a size of a few cm3. The individual measurements are therefore capturing the full three-dimensional shape of the grains. Furthermore, the Fresnel Volume concentrates on a small volume within the sample. The clustering effect due to SIBM-N leads to an over-representation of some clusters within these limited volumes. Thus, these clusters around a few large grains are dominating the measured velocity profile. We have qualitatively analysed this effect and were able to find a combination of two or three clusters that reasonably fits to the actually measured ultrasonic velocity profile. However, several combinations of these four clusters led to similar results and we assume that the fit might also be a statistical effect. We could not find a profound physical explanation. Ultrasonic measurements followed by a COF analysis on the same piece of ice are required to analyse this further. In contrast to the ultrasonic measurements, the thin sections for the COF-derived velocity profiles only provide limited information in the third dimension. This is even more important for an estimated guess of the size of such branched and large grains. Grains close to the thin section but out of plane are invisible for the COF-derived velocity profiles. Furthermore, a cut through a branched large grain may let this grain appear as several small grains, usually called island grains (see Monz et al. (2021), their Fig. 3). A large grain is then underrepresented in the COF-derived profiles, but is more prominent in the Fresnel Volume and therefore more prominent in the velocities measured by the ultrasonic method. This can be regarded as an out–of–plane effect when comparing ultrasonic and COF-derived profiles. To reduce this off-plane effect, we have always combined sets of three thin sections perpendicular to each other (see Hellmann et al. (2021), their Fig. 4) to obtain the COF-derived profiles. As a consequence, the actual number of grains included in the calculations for the COF-derived profiles differs significantly from the number of grains included in the individual ultrasonic measurements, where a few branched large grains may be quite prominent in the actually measured ice volume (see Fig. 5)."

2. *Figure 4 is a very central figure. The data are actually duplicated from a 0-180 to a 0-360 degrees range. I think this might increase artificially the impression of fit and should be avoided. It would be interesting to have the vertical velocity in the same figure, as an extra small column to the right for example, in order to exploit more the vertical direction velocity in the interpretation.*

We adjusted the Figure (0-180° range) and added the vertical measurements to Fig. 4 (and Fig. 6). For clarifying that these measurements contain a periodicity, we have added the first/last measurement to the end/beginning of the profile.

3. *lines 129-130: I do not understand what you are saying here. Your step 4 is a Voigt average (linear average of elastic tensors); when you say here "seismic velocities", do you mean you take the Voigt average (and Reuss and Hill) to calculate the isotropic mean velocities?*

We rephrased this sentence as follows:

The seismic velocities can be calculated from the elasticity tensor or the inverse compliance tensor. Both approaches provide velocity profiles oscillating around an upper (Voigt bound) and lower (Reuss bound) mean velocity (Hill, 1952). We calculated the seismic velocities from both tensors to obtain these upper and lower bounds of the potential velocity range and further derived the velocity profile from the Hill tensor (the mean of elasticity and compliance tensor).

4. *line 163: unclear sentence. The small wavelength does not favour that the individual measurements are a good integrated representation of the whole sample. Do you want to point out here that the wavelength is smaller than the grain size? Anyway, it is only the Fresnel zone dimension that matters to see if the wave field sees one grain at a time along its propagation, not the wavelength.*

We rephrased the sentence for clarification: "Thus, the wavelength is small enough to measure an integrated seismic velocity. This velocity can be regarded as the integrated velocity of the individual grain velocities. Much larger wavelengths may introduce geometric issues such as stationary waves, which are not representative for acoustic waves travelling through the glacier and thus would later inhibit a comparison with in-situ data."

5. *line 186: it seems there are many dark points within the clusters. It is not clear to me why they have been removed.*

Indeed, we excluded all grains below 0.5 mm$^2$ (< 1250 pixels). Those grains usually occur in fissures and as patches within the ice core. However, we reviewed the effect of these small grains and realised that they only minimally affect the calculated velocities (because the grain size is used as weighting factor for the sum over the velocities of the individual grains), visible changes only appear at 22 m and 45 m depth. Therefore, we also include these grains to avoid confusion. We changed Fig 2, 3, 4 and 6 respectively.
During this review, we realised that we used a wrong input file for the velocity calculations in 33 m and corrected this issue (minor changes visible in this figure).

6. *line 228: The coincidence is not as good as stated by this sentence. The maxima for the COF and measured coincide only at depths 2 and 22m. For the three other depths, they do not coincide at all. At 45m, the maximum for the measured coincides with a minimum in COF.*

Together with the new Fig. 4, this discrepancy becomes more obvious and we extended this sentence as follows (to correct for this imprecision):
"All 5 samples show a set of 2 maxima surrounded by 4 minima and 2 local side-maxima. For the samples at 2, 22 and 65 m the positions of the maxima for measured and COF-derived profiles coincide within a range of a few degrees of azimuth (≤15°, Figs. 4a, b, e). At 33 m, there exist a significantly larger azimuthal shift (30°, Fig. 4c) and for the sample at 45 m maxima of one profile coincide with a minimum of the other (Fig. 4d)."

Changes according to Mr. Priors recommendations (Community Comment) and Ms. Maupins review comment to line 218*:*

*about the influence of the core diameter on the anisotropy:*

"Due to the thermal drilling method, thin water layers refroze along the ice core surface, which led to a rough and partially concave surface. This uneven surface and the limited height of the transducer's tip resulted in a poor coupling and we removed the outermost 3 mm thin ice layer (i.e. this

meltwater "skin") by lathing the sample. The ice core diameter was then determined manually for each individual measurement."

Smaller changes according to Ms. Maupins line by line comments:

*line 14: "concise": should be "consistent"?*

Indeed, this should be consistent, changed.

*Figure 2 is cited before Figure 1, as far I can see, and you should normally exchange the figure numbers. As Figure 1 is a good overall summary of your set-up, find a way to cite it before?*

We added a reference to Fig 1c before referencing this Fig 2a-g here.

*line 44-45: rephrase. "since..." does not really make sense with beginning of sentence.*

We rephrased this sentence as follows:
These methods investigate the elastic parameters of the ice. Since elastic parameters and COF are directly related, the methods can also be employed for COF analyses.

*line 59: benchmark to what?*

We replaced this term by "relevant measurement parameter".

*line 101: move sentence to line 128, as this gives the impression you won't give any details, but you give them afterwards, and they are necessary.*

We agree that this sentence may confuse the reader. Therefore, we combined parts of this sentence with text from line 128 and formulated a new sentence in line 129: "The calculations for the polycrystalline tensor and acoustic velocities are described in more detail in Maurel et al. (2015) and Kerch et al. (2018)."

*line 133: at this point you have not said you measure at -5deg. You have said you have frozen the core to -30deg.*

This is true. We rephrased the sentence and added in parentheses that the measured temperature is -5°C as described below. However, at this stage the exact temperatures are not necessary and in addition, they vary between ultrasonic and in-situ experiments.

*line 199: would be good here to have the pure ice value for comparison.*

The given values in this section (incl. Table 1, Fig. 2) are the pure ice values. We clarified this in the captions of Fig 2 and Table 1. We applied the air correction for the first time, when comparing the data with the ultrasonic measurements (Fig. 3).

*line 230: One curve does not look like a smooth version of the other; I do not think you can blame the smoothness for the difference in amplitude.*

We rephrased the respective sentence: "The COF-derived profiles are in general rather levelled with smaller differences between the minima and maxima."

*line 230: This section is about the horizontal velocities, that do not increase with depth. You might remove this sentence.*

We have removed this sentence.

*line 247: you say that the air bubbles not associated with grain boundaries are spherical, but what about the grain boundary bubbles?*

We have added a sub-sentence in parentheses to clarify that bubbles on grain boundaries are influenced by the grain boundary processes and that other processes complicate an interpretation: "(and therefore not pinned to and affected by the boundary pathways)".

We also corrected some typing errors and replaced the reference to Hellmann et al. (2020, preprint in TCD) and Monz et al. (2020, preprint in TCD) with the published versions Hellmann et al. (2021) and Monz et al. (2021).

---

## Author Response (AR2)

**Authors Response:**

*In the response letter to Valerie Maupin, you said that vertical measurements are included in Fig. 4 and Fig. 6, but only Fig. 4 presents the data. Add the same to Fig. 6.*

We have adjusted the Fig. 7 (not Fig. 6, mistake in our RL) respectively.

Other editorial points. All page/line numbers refer the version with tracked changes.

*P3L58: Kluskiewicz et al. (2017, JGlac, 63, 603-617) present sonic logging of WAIS Divide borehole and relevant theories, which might be of your interest.*

Thank you very much! Indeed, a publication that fits perfectly in this content here. Therefore, we adjusted a sentence of our introduction: "Kluskiewicz et al. (2017) have successfully demonstrated the advantages of this method to analyse the COF in ice core boreholes."

*P3L69: typo, "ist" -> "is"*

Changed.

*P3L72: change "CT analyses" to "X ray tomography analyses"*

Changed.

*P3L85: COF is already defined earlier.*

Changed, now only using abbreviation here.

*P6L151: Change to "= 0\deg and 90\deg"*

Changed.

*P8L213: Is it a typo of Fig. 2n?*

Yes, Latex \ref-command adjusted to Fig. 2n.

*P8 Table 1: Clarify that these values are derived from measured COF patterns (not acoustic measurements).*

We adjusted the caption: "Mean, minimum, maximum calculated p-wave velocity (i.e. derived from the COF pattern and not from ultrasonic experiments, without air correction) and degree of anisotropy for each COF sample."

*P9L244: at 45 m*

Changed.

*P10L268: "T" is not defined in this paper. Maybe "ambient temperature of -5oC"?*

We adjusted this information "(\rho = 1.3163 kg\,m^-3 at an ambient temperature of T=-5°C)". Furthermore, we also added "T = -5°C" in line 147 (page 7) to be consistent.

*P10L272: I think citation of Fig. 4 should be changed to Fig. 3.*

No, this reference is correct as we want to refer to the azimuthal profiles and declare that there are no relative changes of these horizontal profiles but just a constant shift of the entire dataset. However, the indirect reference is misleading. To make our point clear we just added an *in* before the reference.

*P17L498-499: remove 2020b reference (it is a duplication of 2021 just below).*

This is only an issue of this track-changes file: Latexdiff compares the old and the new version and then needs both citations to have the old citation printed in red and the new one printed in blue. Unfortunately, the TC template then adds both references. The same appears for Monz et al 2020 and Monz et al 2021 (although they have slightly adjusted their title).
In the final version of the manuscript, it is already correct and only the peer-reviewed papers of 2021 are cited and in the references.

*Figure 3: change the left panel text to "Mean velocity parallel to …."*

Adjusted.

*Figs. 3 and 4: the light red (or pink?) shaded areas are said in different ways to refer the same feature. Please describe it in the same way in these two figures.*

No, the range is slightly different: In Fig. 3, we calculated the mean velocity and the standard deviation of all azimuths in each depth (about 80 measurements). If we would plot maxima and minima, the red area in Fig. 3b would be much larger and any curve would fit since the azimuthal variations are rather large (in contrast, in Fig 3a, no changes would appear since there is no azimuthal variation). In contrast, we want to show the full max-min range of the 4-6 measurements for each azimuth in Fig. 4.

*Figure 4: spell out "respective"*

Changed.

*Figure 6: change the caption to "The mean grain size \phi per sample is…" and remove "and area" from the y axis level.*

Changed.
As announced to the editor, we did not properly update this figure in our previous manuscript version. Now, this issue is also solved. The total number of grains slightly increases due to the adjustments in the clustering algorithm that was requested by Ms. Maupin.